# Anorectal Remodeling in the Transitional Zone with Increased Expression of LGR5, SOX9, SOX2, and Keratin 13 and 5 in a Dextran Sodium Sulfate-Induced Mouse Model of Ulcerative Colitis

**DOI:** 10.3390/ijms252312706

**Published:** 2024-11-26

**Authors:** Mio Kobayashi, Tatsuya Usui, Mohamed Elbadawy, Tetsuhito Kigata, Masahiro Kaneda, Tomoaki Murakami, Takuma Kozono, Yoshiyuki Itoh, Makoto Shibutani, Toshinori Yoshida

**Affiliations:** 1Laboratory of Veterinary Pathology, Cooperative Department of Veterinary Medicine, Tokyo University of Agriculture and Technology, 3-5-8 Saiwai-cho, Fuchu-shi, Tokyo 183-8509, Japan; s212838s@st.go.tuat.ac.jp (M.K.); mshibuta@cc.tuat.ac.jp (M.S.); 2Cooperative Division of Veterinary Sciences, Tokyo University of Agriculture and Technology, 3-5-8 Saiwai-cho, Fuchu-shi, Tokyo 183-8509, Japan; 3Laboratory of Veterinary Pharmacology, Cooperative Department of Veterinary Medicine, Tokyo University of Agriculture and Technology, 3-5-8 Saiwai-cho, Fuchu-shi, Tokyo 183-8509, Japan; fu7085@go.tuat.ac.jp (T.U.); mohamed.elbadawy@fvtm.bu.edu.eg (M.E.); 4Department of Pharmacology, Faculty of Veterinary Medicine, Benha University, Moshtohor, Toukh 13736, Elqaliobiya, Egypt; 5Laboratory of Veterinary Anatomy, Cooperative Department of Veterinary Medicine, Tokyo University of Agriculture and Technology, 3-5-8 Saiwai-cho, Fuchu-shi, Tokyo 183-8509, Japan; fq6451@go.tuat.ac.jp (T.K.); kanedam@cc.tuat.ac.jp (M.K.); 6Laboratory of Veterinary Toxicology, Cooperative Department of Veterinary Medicine, Tokyo University of Agriculture and Technology, 3-5-8 Saiwai-cho, Fuchu-shi, Tokyo 183-8509, Japan; mrkmt@cc.tuat.ac.jp; 7Smart-Core-Facility Promotion Organization, 3-5-8 Saiwai-cho, Fuchu-shi, Tokyo 183-8509, Japan; tkozono@go.tuat.ac.jp (T.K.); yoito@go.tuat.ac.jp (Y.I.)

**Keywords:** transitional zone, LGR5, SOX2, SOX9, ulcerative colitis, mouse

## Abstract

Although hyperplasia of the anorectal transitional zone (TZ) has been reported in mouse models of ulcerative colitis, the mechanisms underlying this phenomenon are not fully understood. We characterized keratin subtypes and examined the expression of stem cell markers in the TZ. Dextran sodium sulfate-treated mice showed abnormal repair of the anorectal region, which consisted of mixed hyperplastic TZ and regenerating crypts. Liquid chromatography-tandem mass spectrometry from the paraffin-embedded TZ in the treated mice revealed that the major keratins were type I cytokeratin (CK)13 and type II CK5, but notable expression of type I CK10 and CK42 and type II CK1, CK4, CK6a, CK8, and CK15 was also detected. Hyperplastic TZ was characterized by the expression of tumor protein 63, sex-determining region Y-box 2 (SOX2), SOX9, and leucine-rich repeat-containing G-protein coupled receptor 5 (Lgr5). Lgr5 was highly expressed in the TZ in the early stages of colitis, followed by higher expression levels of SOX2. The TZ-derived organoids expressed LGR5, SOX2, and SOX9. The present study suggests that transitional zones showing abnormal keratin assembly and stem cell activation may interfere with rectal crypt regeneration, leading to pathological anorectal remodeling in severe colitis.

## 1. Introduction

The transitional zone (TZ) is formed at the boundary between the mucosa and squamous epithelium and has been observed at the cornea–conjunctival [1,2], esophagogastric [3], uterovaginal [4,5], and colon–anal junctions [6,7,8,9]. The TZ consists of a non-keratinizing stratified squamous epithelium in the outermost layers and basal cells in the basal layer to compensate for regeneration [10]. Since TZ cells possess fewer cell junctions, such as desmosomes, they show the potential for migration and invasion. In general, the keratinizing squamous epithelium is more stress-resistant than the mucosal epithelium, as shown in squamous metaplasia [11]. The TZ is more fragile than keratinized squamous epithelium and is more prone to inflammatory reactions, potentially making it the parent tissue for squamous cell carcinoma [1,3,4,5,12]. In studies on the anorectal regions, squamous cell carcinoma and mucoepidermoid carcinoma with squamous metaplasia were found to frequently develop in the perianal and periorbital regions in transforming growth factor (TGF)-β type I receptor conditional-knockout mice [13].

A dextran sulfate sodium (DSS)-induced colitis model can clarify the pathogenesis and therapeutic targets of inflammatory bowel disease, including ulcerative colitis [8,14,15]. Severe colitis is observed in BALB/c mice upon administration of DSS with a molecular weight of 40 kDa, primarily affecting the distal colon [16,17,18]. DSS carries a strong negative charge due to its sulfate groups, making it highly toxic to the colonic epithelial cells and compromising epithelial barrier integrity [19]. Female BALB/c mice are particularly suitable for studying mucosal regeneration, as they demonstrate resilience to severe inflammation and provide an effective platform for analyzing tissue repair mechanisms [20]. This model demonstrates a unique pathological change, i.e., the re-epithelization of squamous epithelium in the erosive rectal and distal colon [8]. The appearance of these lesions indicates irregular regeneration of squamous cells at the edges of the chronic erosions, which may be derived from the TZ in the anorectal junctions. Liu et al. [9] reported the presence of the squamous epithelium, called the squamous neoepithelium of the colon in the anorectal junction, and suggested that the neoepithelium may be derived from tumor protein 63 (p63)-, sex-determining region Y-box 2 (SOX2)-, keratin 14-, and keratin 17-expressing TZ stem cells that can migrate rapidly for colonic wound healing. Furthermore, Mitoyan et al. [7] demonstrated that keratin 17-expressing squamous cell-like TZ cells can become keratin 8-expressing fully differentiated glandular cells, forming crypts with multicellular lineages, such as neuroendocrine cells and goblet cells.

In the present study, we observed hyperplasia of the TZ in a mouse model of DSS-induced colitis. To pathologically characterize the TZ, we dissected the lesions from paraffin blocks to extract the TZ region and attempted to comprehensively identify keratin subtypes using liquid chromatography-tandem mass spectrometry (LC-MS/MS). In addition, to elucidate stem cell lineages and cell proliferation, we determined the expression of the stem cell and cell proliferation marker leucine-rich repeat-containing G-protein coupled receptor 5 (LGR5), SOX2, sex-determining region Y-box 9 (SOX9), p63, and Ki-67, as well as the labeling of 5-bromo-2′-deoxyuridine (BrdU) in hyperplastic TZ. We chose these markers because Lgr5 and SOX9 are expressed in the rectal crypts [21,22]; LGR5, SOX2, and SOX9 are expressed in hair follicles in the anal skin [23,24,25,26], and SOX9 is expressed in the sebaceous glands close to the anal skin [27]. Since the TZ is located between the rectum and anal skin, with hair follicles and perianal sebaceous glands, we hypothesized that LGR5, SOX2, and SOX9 may contribute to the expansion of the TZ during rectal erosion. We also established TZ-derived organoids from DSS-treated mice.

## 2. Results

### 2.1. Hyperplasia of the Anorectal Transitional Zone Is Observed in the Mice with Dextran Sulfate Sodium-Induced Colitis

In control mice, a TZ with non-keratinizing squamous epithelium was observed between the rectum and the anus (Figure 1a). The TZ consisted of multiple layers of basal and squamous epithelial cells without keratohyalin granules or keratinized material. The TZ was connected to the rectal crypts in the proximal region and the anal keratinizing epithelium in the distal region. The anal keratinizing epithelium was also connected to the epidermis with keratohyalin granules and surface keratinization.

In the mice treated with DSS for 6 days (Day 6), crypts were severely atrophied, and the mucosa was mildly to moderately desquamated (ulcerated) (Figure 1a). Consistent with the findings of previous studies [7,8,9], in the rectal mucosa adjacent to the TZ, non-keratinizing squamous epithelial cells extended deep into the lamina propria of the colonic mucosa and toward the proximal colon on the ulcer surface (Figure 1a).

Severe diffuse ulcers were observed in mice treated with DSS for 6 days (Day 6) followed by withdrawal for 6 days (Day 12) (Appendix Aa–c). The ulcers were mainly observed in the rectal mucosa in the TZ between the anus and colon. The non-keratinizing squamous epithelial cells constituting the TZ proliferated, extending toward the ulcerated rectum (Figure 1a). Numerous perianal sebaceous glands were observed beneath the anal epithelium, and the appearance of the glands made it possible to distinguish between the TZ and anus (Appendix Ad). The TZ was significantly longer in the DSS-treated group on Day 12 than in the control and DSS-treated groups on Day 6 (Figure 1b).

In mice treated with DSS for 6 days (Day 6) followed by withdrawal for 20 and 28 days (Days 26 and 34), the ulcers were covered with surface epithelial cells, and short-length regenerative crypts were frequently observed around the ulcers (Figure 1a). The TZ had markedly proliferated in the rectum, showing a disorderly mixture of regenerating crypts and hyperplastic TZ (Appendix A). The TZ infiltrated deep into the lamina propria, and regenerating crypts covered the surface. The smooth muscle layers were irregularly contracted, and isolated or multiple fused lymphoid follicles were formed in the submucosal tissue.

### 2.2. Expression of Cell Adhesion, Cell Proliferation, and DNA Double-Strand Break Markers in the Hyperplastic Anorectal Transitional Zone

We attempted to evaluate whether the TZ had epithelial regenerative capacity by analyzing the expression of E-cadherin, β-catenin, Ki-67, BrdU, p63, and phosphorylation of histone H2AX at serine 139 (γ-H2AX) in the control and DSS-treated mice on days 6 and 12 (Figure 2a and Appendix A). In the control mice, E-cadherin, but not β-catenin, was expressed at cell adhesion in the deep layer of the squamous epithelium. These findings are supported by evidence showing that the anorectal junction possesses fewer desmosomes in mice [6]. Ki-67 was expressed in the basal cells and the deep layer of the squamous epithelium of the TZ, and BrdU labeling also showed a similar pattern when BrdU was injected on days 4 and 5 of DSS administration. p63 was highly expressed in the basal cells and deep layer of the squamous epithelium and mildly expressed in the middle layer of the squamous epithelium of the TZ.

In DSS-treated mice, E-cadherin was expressed at cell adhesion sites in the basal and middle layers of the TZ on Day 6 and was highly expressed in these layers on Day 12 (Figure 2a). β-catenin was not observed on Day 6 but was expressed in the cytoplasm of the TZ cells on Day 12. Ki-67 was expressed in the basal layers of the hyperplastic TZ; however, the labeling indices (LIs) of Ki-67 showed no time course-related changes (Appendix Aa). BrdU was also labeled in the TZ on Day 6, similar to the findings in the control mice, but not on Day 12 (Appendix Ab) when injected on Days 4 and 5 of DSS administration. The epithelial (basal) stem cell marker p63 was consistently expressed in the TZ on Days 6 and 12, similar to that in the control mice (Appendix Ac). These results suggested that hyperplastic TZ cells had similar cell proliferation activity to the control but were completely renewed on Days 6 and 12. Notably, the correlation coefficients between the length of the TZ and the LIs of Ki-67 (R^2^ = 0.9279) (Appendix Ad) and p63 (R^2^ = 0.7554) (Appendix Af) were strong, but the corresponding correlation coefficient for the LI of BrdU was not strong (R^2^ = 0.1807) (Appendix Ae). We further examined the labeling of BrdU analogs in the TZ after injection at 8 h intervals. The results show that the TZ was labeled on Day 6 but not on Day 12 when injected with BrdU analogs on Days 3, 4, 5, and 6 (Figure 2b). However, the TZ was highly labeled when the animals were injected with BrdU analogs on Days 9, 10, 11, and 12 and euthanized on Day 12 (Figure 2b).

The immunohistochemical expression findings for E-cadherin, β-catenin, and Ki-67 on Day 26 are presented in Appendix A. As a DNA double-strand break marker, γ-H2AX was expressed in the TZ in the control and DSS-treated mice; however, no clear changes in expression were detected in both groups (Appendix A).

### 2.3. Stem Cell Markers Are Expressed in the Hyperplastic Anorectal Transitional Zone in the Mice with Dextran Sulfate Sodium-Induced Colitis

Next, we determined the expression of the stem cell markers SOX2, LGR5, and SOX9 in the TZ of control and DSS-treated mice. In the control mice, SOX2 was highly expressed in the basal and middle layers of the TZ (Figure 3a). Lgr5 was not expressed in any layer and SOX9 was weakly expressed in the basal layer. In DSS-treated mice, SOX2 expression in the TZ on Day 6 was less than that in the control group. However, SOX2 was highly expressed in the hyperplastic TZ on Day 12 (Figure 3b). LGR5 was highly expressed in the TZ on Day 6, and its expression on Day 12 was significantly lower than that on Day 6 (Figure 3c). The SOX9 expression pattern in the TZ of the DSS-treated mice was similar to that in the control group (Figure 3d). The correlation coefficients between the length of the TZ and the LIs of SOX2 and Lgr5 were strong (R^2^ = 0.8311 and 07856, respectively) (Figure 3e,f), but the one between the length of the TZ and the LI of SOX9 was weak (R^2^ = 0.3731) (Figure 3g).

Since LGR5, Ki-67, and SOX9 were expressed in both rectal crypts and the TZ (Figure 4a,d,g), we examined the correlation between the regenerative activity of the crypts in the proximal and distal rectum and the TZ. A correlation was detected between regenerative crypts in the proximal rectum and the TZ and the expression rates of LGR5 and Ki-67 (R^2^ = 0.4154 and 0.6491, respectively) (Figure 4b,e). The correlation coefficients for the presence of regenerative crypts in the distal rectum and TZ and the expression rates of LGR5 and Ki-67 were strong (R^2^ = 0.7487 and 0.7763, respectively) (Figure 4c,f). For SOX9, correlations were observed with the regenerative crypts in the proximal and distal rectum and TZ (R^2^ = 0.454 and 0.3906, respectively) (Figure 4h,i).

### 2.4. Establishment of Organoids from the Transitional Zone of Dextran Sulfate Sodium-Treated Mice

We found two types of organoids in the TZ of DSS-treated mice on Day 26. The organoids were composed of non-keratinizing (Figure 5a) and keratinizing squamous cells (Figure 5b). Keratin was observed in the central portion of the organoids along with keratinizing squamous cells. Non-keratinizing organoids expressed Ki-67, SOX2, Lgr5, and SOX9 (Figure 5a), whereas keratinizing organoids expressed only Ki-67, Lgr5, and SOX9 (Figure 5b). E-cadherin, β-catenin, and γ-H2AX were also expressed in both organoids (Figure 5a,b). Non-keratinizing and keratinizing organoids were possibly derived from the TZ and anal skin, respectively.

### 2.5. Co-Localization of SOX2, LGR5, and SOX9 in the TZ

We examined co-localization of SOX2, LGR5, and SOX9 in the TZ on Day 12. SOX2^+^ cells were diffusely expressed in the hyperplastic TZ (Figure 6a–d and Appendix A), and SOX9^high^ cells were distributed in the periphery of the hyperplastic lesions in the lamina propria (Figure 6a,b and Appendix A). Co-localization of SOX2 and SOX9 was observed at the center portion of the hyperplastic lesions of the TZ (Figure 6b,b’). LGR5^+^ cells were also distributed in the periphery of the hyperplastic lesions in the lamina propria (Figure 6c). Some of the LGR5-expressed cells showed co-localization of SOX2 (Figure 6d,d’). Accordingly, SOX2^+^SOX9^low^ cells and SOX2^+^LGR5^low^ cells were identified in the TZ (Figure 6b’,d’).

SOX9^high^ cells and SOX9^low^ cells were observed in the non-keratinizing TZ-derived organoid (Figure 6e,f). Co-localization of SOX2 and SOX9 was confirmed in the organoid (Figure 6f,f’); SOX2^+^SOX9^low^ cells were also identified in the organoid, as seen in the TZ. Co-localization of SOX2 and LGR5 was not clearly shown.

### 2.6. Identification of Keratin 5, 6a, 13, and 15 in the Transitional Zone in the Dextran Sulfate Sodium-Treated Mice

To characterize the keratin phenotype of the TZ, we used LC-MS/MS to identify several types of keratin in the samples dissected from formalin-fixed paraffin-embedded (FFPE) specimens from three control and three DSS-treated mice on Day 12. In control mice, several keratins, namely type I cytokeratin (CK)10 and CK13 and type II CK1, CK5, CK6a, CK6b, CK15, and CK73, were detected in one or two mice, but no CKs were detected in common among the three mice (Table 1). In DSS-treated mice, type I CK13 and type II CK5 were detected in all three mice on Day 12. Type I CK10 and CK42 and type II CK1, CK4, CK6a, CK8, and CK15 were detected in one or two mice. The scores of CK10 and CK1 (>1000) were higher than those of the others and were correlated with the higher exponentially modified protein abundance index (emPAI) values in the DSS-treated mice. Type I CK13 and type II CK5 were commonly detected in the anal squamous epithelium in control and DSS-treated mice (Appendix A). Type I CK10 and CK42 and type II CK1, CK4, CK6a, CK8, CK14, and CK15 were detected in both groups of mice.

## 3. Discussion

The TZ at the anorectal junction occupies a unique stem cell niche between the rectum and anus. We incidentally observed a hyperplastic TZ in DSS-induced colitis [28], and these lesions were consistent with previously reported findings [8]. Some researchers have also reported the proliferation of TZ cells in DSS-induced colitis and other models [6,7,8,9,10,11]. We aimed to explore the mechanism underlying this unique response in the anorectal junction. Previous studies have shown that the lesions may be regulated by the keratin 17-positive stem cell population [7,12], which was labeled with the stem cell markers p63 [9], SOX2 [9,10], and CD34 [6,7,10]. The present study further explored the keratin and stem cell phenotypes in the TZ of DSS-treated mice. The main findings of the present study are as follows: (1) we characterized the keratin subtypes in the TZ in the DSS-induced colitis model using LC-MS/MS of FFPE specimens; (2) we identified the time course of the expression of the stem cell markers Lgr5, SOX2, and SOX9 together, even with similar cell proliferation activities as the control, and correlated the expression of Lgr5 and SOX9 in the regenerative crypts and hyperplastic TZ; and (3) we also established a possible TZ-derived organoid *in vitro*, which is the first description of this organoid in the literature, to the best of our knowledge.

LC-MS/MS of FFPE specimens is a powerful tool for identifying specific proteins in the local regions of HE-stained sections [29,30,31]. We identified CK5 and CK13 in the TZ of all three mice treated with DSS for six days followed by withdrawal for six days (Day 12). These results suggest that CK5 and CK13 may be the principal keratins in the hyperplastic TZ. CK5 is a basic keratin (type II) which is distributed in the basal cells of the stratified epidermis and is correlated with the mitotic activity of epithelial cells [32,33]. CK13 is an acidic keratin (type I) that is distributed in the suprabasal cells of the mucosal epithelia. CK5 and CK13 are located together with the partners CK14 and CK4 to form heterodimers, respectively [32]; however, the partner keratins were identified in only one mouse in the present study. These results suggest that the hyperplastic TZ may be composed of dysregulated keratin subtypes. This hypothesis was supported by the detection of several non-orchestrated keratins: type I CK10 and CK42 and type II CK1, CK4, CK6a, CK8, and CK15. Among them, CK10 and CK1 may be essential for understanding TZ biology, since they had the highest scores (qualitative parameters) and emPAI values (quantitative parameters) in the LC-MS/MS analysis in the present study. CK10 is expressed in the post-mitotic suprabasal layers of the epidermis and prevents further cell division and differentiation into keratinocytes [32]. Interestingly, CK10 has been suggested to form heterodimers with CK1 and integrate into the keratin filament network. In previous studies, Western blotting and fluorescence immunohistochemical analyses have demonstrated the expression of CK5, CK7, CK8, CK14, CK17, and/or CK20 in the TZ of mouse models [6,7,9]. CK17-positive keratinocytes exhibit multipotency at the anorectal junction [7,9]. Patients with ulcerative colitis-associated neoplasia (UCAN) show extension of the squamous epithelium from the anus in endoscopic assessments and a border between the p63^+^CK5^+^CK17^+^ transitional squamous epithelium and the p53^+^ proliferative dysplastic epithelium in pathological assessments [12]. However, we could not detect CK17 in the present study. This discrepancy may be attributable to the small population of CK17-positive stem cells in the hyperplastic TZ. Accordingly, hyperplasia of the TZ indicated dysregulated keratin expression in wound healing and cancer properties in the mouse model, and further studies are required to understand the role of keratin dysregulation in the pathological remodeling in the TZ.

Previous studies on the TZ of normal or rectally wounded mice have shown that overexpression of the stem cell markers SOX2, p63, and CD44 acts directly on cell proliferation [6,7,10]. In a study with the DSS mouse model, Liu et al. reported that the Sox2^+^ Krt14^+^ Krt7^+^ squamous cells in the distal TZ were called the squamous neoepithelium of the colon (SNEC) and represented a small but specialized population [9]. Our data demonstrate that a combination of Lgr5 and SOX2 may regulate the proliferation of TZ cells in the mouse model. Lgr5^+^ stem cells are typical stem cells located in the crypt base of the colon as well as in the bulge of the hair follicle [25]. Upon wound healing, Lgr5^+^ stem cells generate new hair follicles and appear in the interfollicular epidermis [25]. However, to the best of our knowledge, the role of Lgr5 in TZ cell proliferation has not yet been investigated. SOX2^+^ stem cells are located in the bulge of hair follicles and regulate hair follicle development [23,24]. Similar to Lgr5, SOX2 promotes epidermal wound healing *in vivo* and *in vitro* [34]. Our data suggest that SOX2 expression following Lgr5 expression may promote TZ hyperplasia. This hypothesis is partially supported by the finding showing that SOX2, in combination with p63, maintains and promotes TZ remodeling, as in squamous metaplasia [9,10], which is also regulated by SOX2 and p63 [11]. Since SOX2 induces angiogenesis in interstitial tissues during wound healing [34], this transcription factor may heal severe ulcers by targeting multicellular lineages in the anorectal junctions. SOX9 has multiple potential functions as a progenitor cell in hair follicles and regulates sebaceous gland morphogenesis, melanocyte differentiation [26,27], and corneal epithelial healing [35]. Although the function of each gene in hair follicles and the epidermis has been reported, the interactions of SOX2, SOX9, and Lgr5 in wound healing have not been fully investigated. In the colon, Lgr5 and SOX9 actively control the proliferation of stem cells and retain intestinal stem cells [21,22]. Although no time course changes were observed in SOX9 expression in the TZ, the multipotential SOX family transcription factor may play a role in TZ proliferation by coordinating with Lgr5 and Sox2.

The development of squamous cell carcinoma in the rectum-anal region has been reported in patients with UCAN [12], but the underlying mechanism has not been thoroughly investigated. Most rectal and anal cancers are adenocarcinomas and squamous cell carcinomas, respectively [36]. Lesions with dysplasia and squamous metaplasia have been suggested to be precursors of UCAN [12]. Studies on the TZ in mouse models have also raised questions about the potential for the formation of squamous cell carcinoma [6,7,8,9,10,12]. The present study demonstrated hyperplasia of the TZ in a DSS-induced colitis model; however, no evidence of cancer transformation was observed. SOX2 and SOX9 regulate skin tumors, including squamous cell carcinoma [37,38]. In addition, the effects of the correlation between SOX2 and SOX9 expression in malignant tumors, such as lung cancer and breast cancer, have attracted attention [39]. TGF-β signaling is one of the major pathways controlled by the inverse expression of SOX2-SOX9 [40]. A conditional TGF-β receptor II knockout mouse model demonstrated spontaneous development of squamous cell carcinoma at the junction of the mucosal and stratified squamous epithelium of the anal canal, with TGF-β receptor II deficiency observed in the simple layer of the anal column but not in the adjacent skin [13]. This mouse model may demonstrate the importance of the TGF-β pathway in maintaining tissue homeostasis in the TZ, but no previous reports have investigated the relationship between TGF-β and SOX2-SOX9. However, the role of Lgr5 in squamous cell carcinoma remains a topic of debate. Lgr5 is highly expressed in squamous cell carcinomas in the oral cavity and esophagus and may play a role in cancer stem cell properties [41]. Conversely, in the TZ of the gastroesophageal junction, Lgr5-expressing cells can transform into Lgr5-non-expressing progenitor cells that may develop into dysplasia and squamous cell carcinoma [42]. Lgr5 is not distributed in chemical- and ultraviolet-induced epidermal tumors [25], nor does it participate in tumor maintenance [43]. Our data suggest that the lesions of the TZ with Lgr5 expression might be remodeled to severe colitis, but they did not have the potential for squamous cell carcinoma under the present study conditions.

A previous study has demonstrated that slow-cycling BrdU-retaining cells are a putative stem cell population in the TZ [6]. In the stratified epithelium of the anal canal and the intestinal-type glandular mucosa of the anal TZ in normal mice, all epithelial cells retained the BrdU label. After 26 days, label-retaining cells were expressed in the basal layer of the anus [10]. However, these findings are inconsistent with the results of our study, which found that BrdU-labeled cells were completely lost on Day 12 when BrdU was injected twice on Days 3 and 4. This discrepancy can be attributed to the pathobiological condition of the TZ; the normal TZ maintains the stem cell population for a long time [6], but upon healing, as in our study, the TZ enhances cell renewal. Cell proliferation in the basal layer of the TZ was consistently maintained under normal conditions and during wound healing in colitis, as shown by Ki-67 labeling. However, we observed markedly high cell proliferation activity on Day 12 when BrdU analogs were injected during Days 9 and 12. The TZ in abnormal pathological remodeling might be provided by cumulative cell proliferation due to Lgr5-maintained stem cell activation to heal severe ulcers in the rectum during an emergency.

We selected female Balb/c mice for this study, as they can recover from mucosal injury more effectively than males, making them ideal for observing mucosal healing [20]. This facilitated precise observation of TZ hyperplasia in the present study. Previous research has shown that the activation of the estrogen receptor (ER) in female mice exacerbated acute DSS-induced colitis, while inhibition of ERα may reduce inflammation, indicating that ovarian hormones such as estradiol may significantly contribute to inflammation [44]. However, studies specifically investigating the role of ER signaling in TZ hyperplasia remain limited. Existing research on TZ hyperplasia has utilized both male and female mice [7,9], and the impact of sex differences on TZ response remains unknown. Hormonal changes associated with aging may significantly alter inflammatory and regenerative responses. Further research is required to investigate the role of estrogen signaling and TZ responses, including detailed comparisons of sexes and age groups, including young adult and post-reproductive (retired) female mice with the inactivation of estrogen signaling, potentially enhancing the translational relevance of the model.

Two types of organoids, namely those of mucosal epithelial and squamous epithelial origin, were observed among the organoids prepared from the anorectal region. Organoids derived from the squamous epithelium were round with internal keratinization, similar to the keratinized squamous epithelium of the anus. Organoids derived from the TZ showed no keratinization and had a protruding limbus-like shape, similar to the crypt-like proliferation induced by DSS administration. To the best of our knowledge, this is the first report describing a TZ-derived organoid that might be regulated with SOX2, SOX9, and LGR5. We demonstrated SOX2^+^SOX9^low^ cells in the TZ and its derived organoid. SOX9 is essential for follicular and sebaceous glands [26,27]; however, the expression levels of SOX9 have not been elucidated in these tissues. In the intestine, SOX9 is highly expressed in the crypt case, gradually decreasing in the upper sites of the crypts, the transit-amplifying zone [21]. SOX9^high^ and SOX9^low^ crypt cells were proposed as reserve and active interstitial stem cells, respectively [22]. SOX9^low^ crypt cells can contribute to mucosal regeneration and form organoids in radiation-mediated mucosal injury [45]. SOX9^high^ cells might accelerate the proliferation of the TZ at the periphery portions, like crypt bases, and SOX2^+^SOX9^low^ cells might maintain the TZ at the center portions. Further studies are required to investigate the interaction of the stem cells and keratin subtypes, some of which are distributed in the basal and suprabasal layers of stratified epithelia, plausibly regulating stemness and differentiation of the TZ.

Taken together, our results suggest that the TZ in the DSS model shows abnormal keratin regulation, leading to abnormal pathological remodeling in anorectal regions with severe erosion. This response may be regulated by the expression of typical stem cell populations that are commonly expressed in both the rectum and anal skin, thereby maintaining an abnormal balance between mucosal regeneration and squamous metaplasia. These responses may rescue severe ulcers in the distal colon; however, therapeutic approaches are required to control both cell responses and reduce TZ remodeling, which may increase the risk of squamous cell carcinoma depending on the long-term course of the disease. TZ-derived organoids may help identify therapeutic targets to reduce remodeling at the site.

## 4. Materials and Methods

### 4.1. Chemicals

DSS (molecular weight 36,000–50,000, CAS number 9011-18-1) was purchased from MP Biomedicals (Santa Ana, CA, USA). Five-bromo-2′-deoxyuridine (BrdU, molecular weight 307.10, CAS number 59-14-3), 5-Iodo-2′-deoxyuridine (IdU, molecular weight 354.10, CAS number 54-42-2), and 5-Chloro-2′-deoxyuridine (CldU, molecular weight 262.65, CAS number 50-90-8) were purchased from Sigma-Aldrich (St. Louis, MO, USA), Tokyo Chemical Industry Co., Ltd. (Chuo-ku, Tokyo, Japan), and Sigma-Aldrich (St. Louis, MO, USA), respectively.

### 4.2. Animals

Four-week-old female BALB/cAnNCrlCrlj mice (Charles River Laboratories Japan, Inc., Kanagawa, Japan) were acclimated for nine days to the testing environment under controlled temperature (23 ± 3 °C), humidity (5 ± 20%), and light (12:12 h light–dark cycle) conditions. Three to four mice were maintained in individual plastic cages with enrichment and safe harbors and were provided free access to a basal diet (Oriental MF, Oriental Yeast, Tokyo, Japan) and tap water. Food consumption and water intake were measured twice a week per cage, and body weight was measured daily during the study. All procedures in this study were conducted in accordance with the Guidelines for Proper Conduct of Animal Experiments (Science Council of Japan, 1 June 2006), and the protocols were approved by the Animal Care and Use Committee of the Tokyo University of Agriculture and Technology (Fuchu, Tokyo, Japan). The studies were performed in compliance with the ARRIVE guidelines.

### 4.3. Experimental Design

Experiment I

The first experiment was conducted as described previously [28]. We used paraffin-embedded colorectal samples for histopathological, immunohistochemical, and LC-MS/MS analyses of the TZ. Briefly, negative controls (untreated mice) received sterile distilled water during the study (n = 4). The treated mice received 5% (*w*/*v*) DSS dissolved in sterile distilled water as drinking water for six days (Day 6), followed by withdrawal of DSS for six days (Day 12) to induce colitis. The mice were anesthetized on Days 6 (control mice, n = 2; DSS-treated mice, n = 11) and 12 (control mice, n = 2; DSS-treated mice, n = 10) by isoflurane inhalation, euthanized via exsanguination from the abdominal aorta and vena cava, and necropsied.

Experiment II

The mice (n = 16) were divided into two groups, a negative control group (n = 4) and a DSS-treated group (n = 12), on the basis of their body weight at the initiation of the study. Negative controls (untreated mice) were administered sterile distilled water. In the DSS-treated group, the mice received 4% (*w*/*v*) DSS dissolved in sterile distilled water as drinking water for 6 days (Day 6), followed by the withdrawal of DSS for 6 days (Day 12) to induce colitis (n = 6), as reported previously [28,46,47]. The remaining DSS-treated mice (n = 6) were subjected to DSS withdrawal for 20 days (Day 26). The mice were anesthetized on Days 12 and 26 (control mice, n = 2; DSS-treated mice, n = 6 at each point) by isoflurane inhalation, euthanized via exsanguination from the abdominal aorta and vena cava, and necropsied. One animal from each control and treated group was used for organoid preparation on Day 26.

Experiment III

The mice (n = 18) were divided into two groups on the basis of their body weight at the initiation of the study: a negative control group (n = 6) and a DSS-treated group (n = 12). The mice were treated as described in Experiment II. A cell proliferation indicator (BrdU) was administered intraperitoneally (100 mg/kg; twice every 24 h) on Days 4 and 5 of DSS administration, as described previously [46,47]. The mice were anesthetized on Days 6 (control mice, n = 2; DSS-treated mice, n = 4), 12 (control mice, n = 2; DSS-treated mice, n = 4), and 34 (control mice, n = 2; DSS-treated mice, n = 4) by isoflurane inhalation, euthanized by exsanguination from the abdominal aorta and vena cava, and subsequently necropsied.

Experiment IV

The mice (n = 12) were divided into two groups according to their body weight at the initiation of the study: a negative control group (n = 4) and a DSS-treated group (n = 8). The mice were treated as described in Experiment II. The cell proliferation indicators 5-chloro-2′-deoxyuridine (CldU), BrdU, and 5-Iodo-2′-deoxyuridine (IdU) were administered intraperitoneally to the mice (n = 2 for the control; n = 4 for the treated group) every 8 h on Days 3, 4, 5, and 6, and the mice were then euthanized on Day 6 before the last 1 h injection. The remaining mice (n = 2 for the control group; n = 4 for the treated group) received these analogs on Days 9, 10, 11, and 12 and were euthanized on Day 12 before the last 1 h injection. The dose (3.3 mmol) and 8 h pulse injection were based on previously reported methods [48,49,50].

### 4.4. Disease Activity Index

The disease activity index (DAI) was based on body weight loss, diarrhea, and fecal blood, as described previously in Experiment I [28]. The DAI in Experiments I–III is shown in Appendix A, respectively.

### 4.5. Tissue Preparation and Histological Evaluation

The colorectum was removed from the large intestine, and its length was measured [23]. The colorectum was divided into the distal and proximal regions and was opened longitudinally, the mucosal surface was exposed, and the tissue was sandwiched between filter paper and fixed in 4% paraformaldehyde in phosphate-buffered saline (PBS) for several hours. After fixation, the tissue was cut in half and embedded in paraffin with the mucosal surface being placed inward. The paraffin-embedded blocks were sectioned and subjected to hematoxylin and eosin (H&E) staining for morphology evaluation and measurement of TZ length. The histological analysis was performed using a vertical slide system (VS120, Olympus Corporation, Tokyo, Japan).

### 4.6. Immunohistochemical Analysis

Immunostaining was performed manually using antibodies against E-cadherin, β-catenin, Ki-67, p63, BrdU, SOX2, LGR5, SOX9, and γ-H2AX (Appendix A). The samples were incubated with the primary antibodies overnight at 4 °C. An avidin-biotin-peroxidase complex method was employed using the VECTASTAIN ^®^ EliteABC Kit (Vector Laboratories Inc., Burlingame, CA, USA) with 3,3′-diaminobenzidine as a chromogen, followed by counterstaining with hematoxylin for microscopic examination. In the internal controls, positive signals for Lgr5, SOX9, and Ki67 were detected in the crypt, but the signal for SOX2 was negative (Appendix A); positive signals for Lgr5, Ki-67, p63, and BrdU were detected in the follicles, but the signal for SOX2 was completely negative (Appendix A); and positive signals for p63, SOX9, E-cadherin, and β-catenin were detected in the sebaceous glands, but the signals for Lgr5, SOX2, and Ki-67 were negative (Appendix A).

The expression patterns of immunopositive reactions in the anorectal transition zone were analyzed quantitatively using a vertical slide system (VS120, Olympus Corporation). The ratio of positive cells to non-keratinized squamous epithelial cells that constituted the TZ per total number of cells was calculated, and the correlation between positive cells and the total length of the TZ was analyzed. Furthermore, the correlation between the positive cells in the TZ and regenerative crypts in the distal colon was calculated. The number of positive cells in the TZ and crypts was determined using image analysis software Fiji (https://imagej.net/software/fiji/downloads), accessed on 1 October 2021 (Appendix A).

### 4.7. Immunofluorescence and Confocal Microscopy

The colon tissue and organoids were fixed with 4% paraformaldehyde for 24 h at 4 degrees. Subsequently, they were treated with conventional methods and stained with the following antibodies: SOX2 (1:200; Abcam, ab79351), SOX9 (1:250; Abcam, ab185966), LGR5 (1:400; Abcam, ab219107), (Abcam, Cambridge, UK), Goat Anti-Rabbit IgG H&L (Alexa Fluor^®^ 488; 1:1000; Abcam, ab150077), and Goat Anti-Mouse IgG H&L (Alexa Fluor^®^ 555; 1:1000; Abcam, ab150114). Nuclei were stained with ProLong™ Gold Antifade Mountant with DNA Stain DAPI (Invitrogen, Waltham, MA, USA, P36930). Images were obtained with confocal microscopy (Nikon, AX R, Japan). The confocal microscopy imaging was performed at Tokyo University of Agriculture and Technology for Smart-Core-Facility Promotion Organization.

### 4.8. Proteomics Analysis

The TZ areas were collected from FFPE samples of the anorectum from three control and three DSS-treated mice (Day 12), and the constituent proteins were detected using mass spectrometry (MS)-based proteome analysis as described previously [51], with minor modifications. Each FFPE specimen was cut into 7 μm sections on polyethylene naphthalate membrane glass slides (Thermo Fisher Scientific, Waltham, MA, USA) and stained with H&E. The TZ lesions (159,274.42–275,700.94 μm^2^) were collected under stereomicroscopy (SMZ745; Nikon, Tokyo, Japan) (Appendix A). The collected samples were immersed in lysis buffer (10 mM Tris, 1 mM ethylenediaminetetraacetic acid, and 0.002% Zwittergent 3–16; Calbiochem, San Diego, CA, USA), boiled for 120 min, and sonicated for 60 min. Each sample was digested with trypsin (mass spectrometry-grade; Fujifilm Wako Pure Chemicals, Osaka, Japan), followed by reduction with dithiothreitol (Fujifilm Wako Pure Chemicals). LC-MS/MS analysis of the digested peptides was performed using a high-performance liquid chromatograph coupled with a mass spectrometer (LTQ Orbitrap XL; Thermo Fisher Scientific). The obtained tandem mass spectrometry data were used to calculate the theoretical fragmentation pattern of trypsin-digested peptides using the Mascot Server (Version 2.4, Matrix Science Ltd., London, UK) and were then collated to the mouse protein data against the combined database of UniProt_Mouse using the downloaded data for 63,447 sequences (UniProtKB/Swiss-Prot: UP000000589). Qualitative and quantitative data were presented as scores and emPAIs, respectively [51]. The proteomics analyses were performed at Tokyo University of Agriculture and Technology for Smart-Core-Facility Promotion Organization.

### 4.9. Organoids

The distal regions from one control and one DSS-treated mouse were temporarily stored in a stock solution for organoid preparation [52]. The tissues were squeezed into a dish containing 4 mL of PBS with scissors. After washing, the PBS was aspirated with an aspirator, and the process was repeated three times. The tissues were gathered by tipping them into a dish with 5 mL of 10× advanced Dulbecco’s modified Eagle medium (DMEM; FBS-free, final concentration 0.125 mg/mL; Liberase TH; 5401135001, Sigma-Aldrich, St. Louis, MO, USA) and 10× solution (1.25 mg/mL, dissolved with advanced DMEM containing 1% PS) and then transferring them to one containing 0.5 mL + 4.5 mL of advanced DMEM with 1% PS. The tubes were incubated in a 37 °C water bath for 30 min with shaking by pipetting with a 1 mL pipette every 15 min. The solutions were observed directly under a microscope to observe the epithelial cells. Then, the solutions were passed through a 100 μm cell strainer and transferred to a 15 mL tube. The tubes were centrifuged at 600× *g* (2000 rpm) for 5 min, and the supernatant was removed. PBS (8 mL) was added to the tubes, followed by pipetting up and down and centrifugation at 600× *g* (2000 rpm) for 5 min. The precipitates were washed with PBS three times, and the supernatant was removed. The cells were resuspended in Matrigel on 24 well plates (40 µL/well) and kept in an incubator for 30 min. Next, 500 mL of WENR medium containing B-27+Y+ primocin was added. At the beginning of the culture, CHIR was added to the medium at a dilution of 1:1000. The culture medium was changed three times per week. When the cells reached confluence, they were processed for histopathological and immunohistochemical analyses.

### 4.10. Statistical Analysis

The Tukey–Kramer test or Steel–Dwass test was used for quantitative analysis of the length of the TZ, LIs of cell proliferation, and the expression data for stem cell markers in the control and DSS-treated groups (Days 6, 12, 26, and 34). Different letters were used to indicate significant differences between the groups (*p* < 0.05). We also obtained correlation coefficients between the length of the TZ and the LIs of cell proliferation and stem cell markers, and the number of positive cells for cell proliferation and stem cell markers in the TZ and crypts.

## Figures and Tables

**Figure 1 ijms-25-12706-f001:**
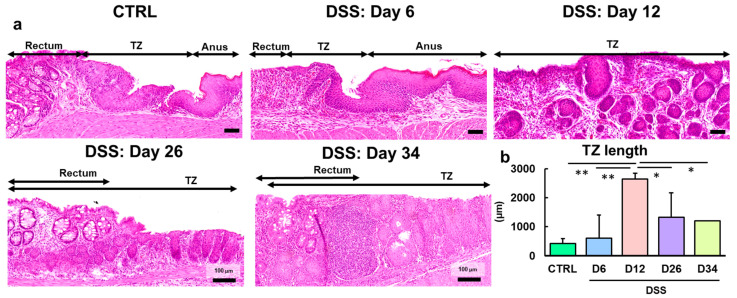
Morphological characterization of the TZ in the control and DSS-treated groups. (**a**) Representative images of the TZ in the control mice and mice administered 4% or 5% DSS in drinking water for 6 days (Day 6), followed by withdrawal of DSS for 6 days (Day 12), 20 days (Day 26), and 28 days (Day 34). In the control mice, the TZ with non-keratinizing squamous epithelium was observed between the rectum with crypts and the anus with keratinizing squamous epithelium. In the DSS-treated mice, the non-keratinizing squamous epithelium covered the ulcer lesions in the rectum on Day 6 and proliferated intensely toward the depths on Day 12. On Days 26 and 34, pathological remodeling was shown as mixed with regenerative crypts (light blue triangles) and the hyperplastic TZ (yellow triangles) in the ulcer regions (see Appendix A). Hematoxylin and eosin stain. Bar = 50 (CTRL, Days 6 and 12) or 100 µm (Days 26 and 34). (**b**) Comparison of TZ lengths in the control and DSS-treated groups at each time point. The mice were treated with 5% DSS for 6 days and euthanized on days 6 and 12. The other mice were treated with 4% DSS for 6 days and euthanized on days 26 and 34. N = 3 (CTRL), 6 (D6), 9 (D12), 3 (D26), and 2 (D34). *, ** Significant differences between each time point or region (*p* < 0.05 or 0.01, Tukey–Kramer multiple comparison test). CTRL, control; DSS, dextran sodium sulfate; D6, Day 6; D12, Day 12; D26, Day 26; D34, Day 34; TZ, transitional zone.

**Figure 2 ijms-25-12706-f002:**
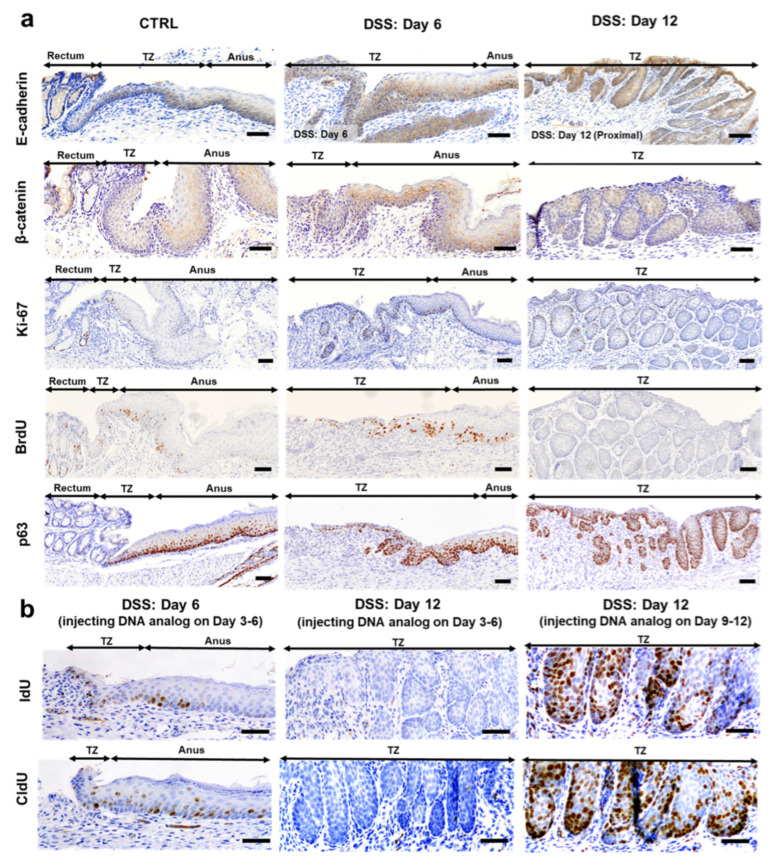
Immunohistochemical analyses of cell adhesion, cell proliferation, and epithelial stem (basal) cell markers in the TZ in the control and DSS-treated groups. (**a**) Representative images of the expression of E-cadherin, β-catenin, Ki-67, and p63 and BrdU labeling in the control mice and mice administered 5% DSS in drinking water for 6 days (Day 6) followed by withdrawal of DSS for 6 days (Day 12). Cell adhesion proteins E-cadherin and β-catenin are less or weakly expressed in comparison with the expression levels in the anus. Ki-67 is expressed in the basal layers of crypts, TZ, and anus, and BrdU is labeled in the control and DSS-treated groups on Day 6, but not on Day 12, when injected twice on days 4 and 5 (see text Experiment III for BrdU labeling). p63 is highly expressed in the basal and suprabasal layers in the TZ and anus, but not in the crypts. The positive signals are visualized with 3,3′-diaminobenzidine as a chromogen (brown), followed by counterstaining with hematoxylin. Bar = 50 µm. (**b**) IdU or CldU is labeled in the TZ on Day 6, but not on Day 12, when injected intraperitoneally at 8 h intervals from Days 3 to 6. Both analogs are labeled in the TZ on Day 12 when injected intraperitoneally at 8 h intervals from Days 9 to 12 (see text Experiment IV for CldU, BrdU, and IdU labeling). BrdU, 5-bromo-2′-deoxyuridine; CldU, 5-chloro-2′-deoxyuridine; IdU, 5-Iodo-2′-deoxyuridine; DSS, dextran sodium sulfate; p63, tumor protein 63; TZ, transitional zone.

**Figure 3 ijms-25-12706-f003:**
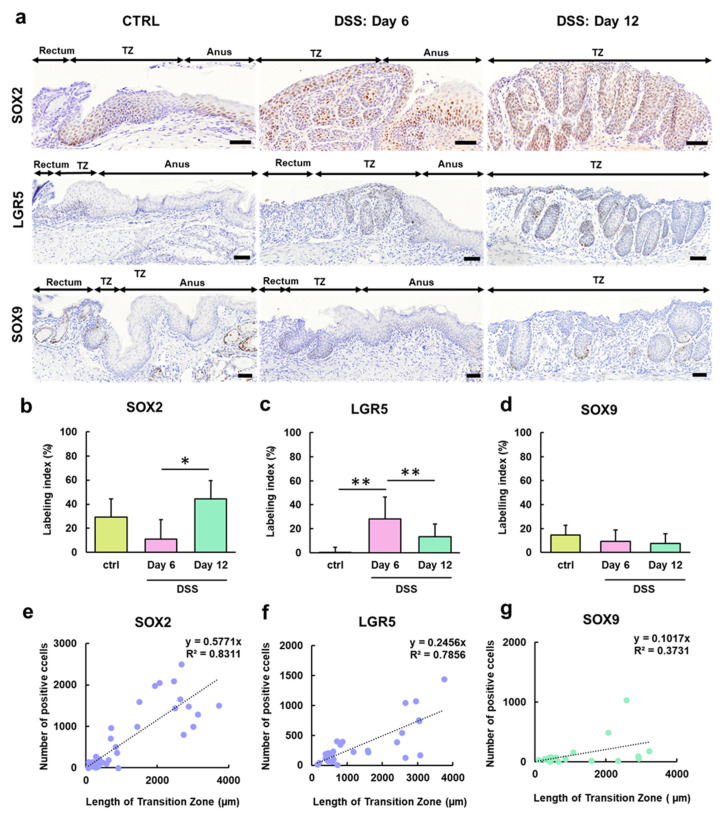
Immunohistochemical analysis of stem cell markers in the TZ in the control and DSS-treated groups. (**a**) Representative images of the expression of SOX2, LGR5, and SOX9 in the control mice and mice administered 5% DSS in drinking water for 6 days (Day 6) followed by withdrawal of DSS for 6 days (Day 12). SOX2 is highly expressed in the TZ and anus. Labeling indices of SOX2 (**b**), LGR5 (**c**), and SOX9 (**d**) in the TZ. Scatter plots with correlation coefficient (R^2^) and linear function (y = ax) between the number of cells showing positive results for SOX2 (**e**), Lgr5 (**f**), and SOX9 (**g**) and the length of the TZ. *, ** Significant differences between each time point or region (*p* < 0.05 or 0.01, Tukey–Kramer multiple comparison test). DSS, dextran sodium sulfate; LGR5, leucine-rich repeat-containing G-protein-coupled receptor 5; SOX2, sex-determining region on Y-box transcription factor 2; SOX9, sex-determining region on Y-box transcription factor 9; TZ, transitional zone.

**Figure 4 ijms-25-12706-f004:**
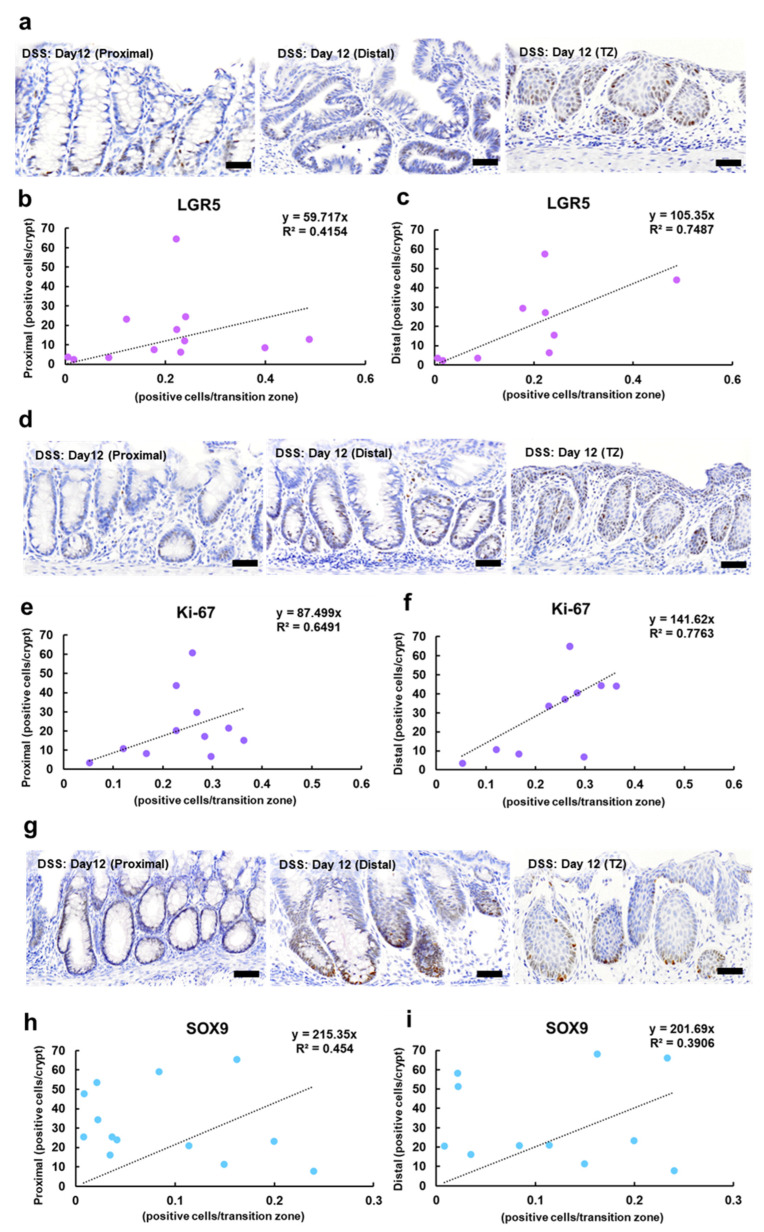
Immunohistochemical analysis of stem cell and cell proliferation markers in the crypts and TZ in the DSS-treated mice. Representative images showing nuclear expression of LGR5 (**a**), Ki-67 (**d**), and SOX9 (**g**) in the TZ of the treated mice administered 5% DSS in drinking water for 6 days (Day 6), followed by withdrawal of DSS for 6 days (Day 12). The positive signals are visualized with 3,3′-diaminobenzidine as the chromogen (brown), followed by counterstaining with hematoxylin. Bar = 50 µm. Scatter plots with correlation coefficient (R^2^) and linear function (y = ax) between the number of positive cells in the crypts and TZ in the labeling of LGR5, proximal colon (**b**), distal colon (**c**); Ki-67, proximal colon (**e**), distal colon (**f**); and SOX9, proximal colon (**h**), distal colon (**i**). DSS, dextran sodium sulfate; LGR5, leucine-rich repeat-containing G-protein-coupled receptor 5; SOX9, sex-determining region on Y-box transcription factor 9; TZ, transitional zone.

**Figure 5 ijms-25-12706-f005:**
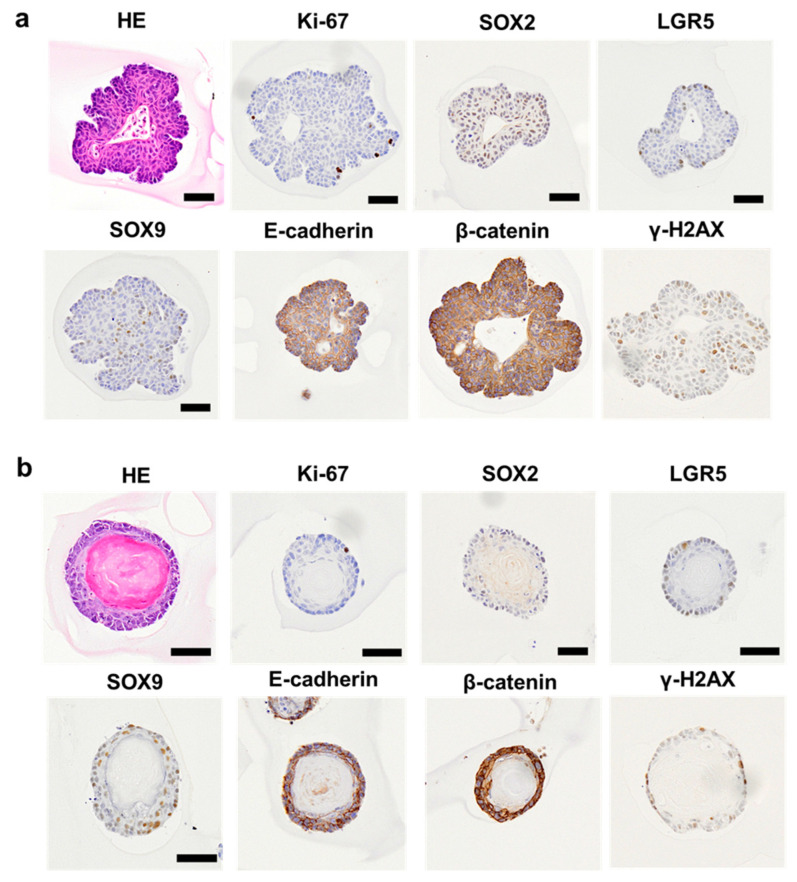
Morphological and immunohistochemical characterization of TZ- and anus-derived organoids. Representative images of organoids of non-keratinizing (**a**) and keratinizing squamous cells (**b**) in mice administered 4% DSS in drinking water for 6 days followed by the withdrawal of DSS for 21 days (Day 26). Organoids with non-keratinizing and keratinizing squamous cells were derived from the TZ and anus, respectively. Hematoxylin and eosin staining. Bar = 50 µm. Immunohistochemical expression of Ki-67, SOX2, LGR5, SOX9, E-cadherin, β-catenin, and γ-H2AX in TZ- (**a**) and anus-derived organoids (**b**). The positive signals are visualized with 3,3′-diaminobenzidine as a chromogen (brown), followed by counterstaining with hematoxylin. Bar = 50 µm. DSS, dextran sodium sulfate; γ-H2AX, phosphorylation of histone H2AX at serine 139; LGR5, leucine-rich repeat-containing G-protein-coupled receptor 5; SOX2, sex-determining region on Y-box transcription factor 2; SOX9, sex-determining region on Y-box transcription factor 9; TZ, transitional zone.

**Figure 6 ijms-25-12706-f006:**
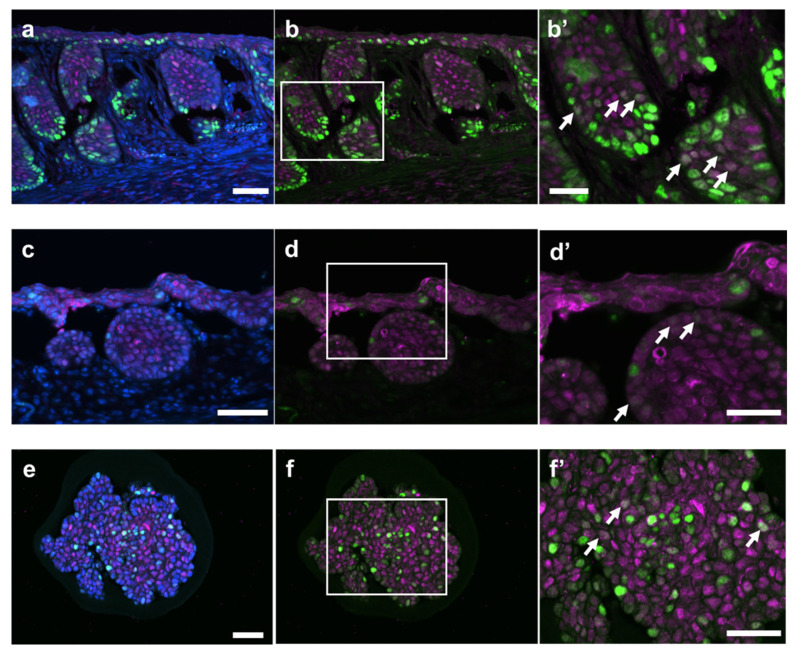
Some SOX2-retaining cells colocalize with SOX9 or LGR5-retaining cells. Representative images of TZ (**a**–**d**) and organoids of a non-keratinizing squamous cell (**e**,**f**) in the treated mouse administered 4% DSS in drinking water for 6 days, followed by withdrawal of DSS for 6 days (Day 12: (**a**–**d**) and 21 days (Day 26; (**e**,**f**)). Double immunofluorescence staining for SOX2 (Magenta) and SOX9 (Green) (**a**,**b**) or SOX2 (Magenta) and LGR5 (Green) (**c**,**d**) in TZ. Double immunofluorescence staining for SOX2 (Magenta) and SOX9 (Green) in the organoid (**e**,**f**). Nuclei were counterstained with DAPI (blue). Higher magnification of the region for indicated square (white line) in Figure 6b,d,f), with a white arrow showing colocalization (**b’**,**d’**,**f’**). SOX2 immunoreactivities in mice and organoids derived from TZ were observed in the nucleus as a diffuse pattern. LGR5 and SOX9 were mainly observed in the nucleus for a marginal region at TZ. The nuclei colocalized with SOX2 and SOX9 (SOX2^+^SOX9^low^ cells) or SOX2 and LGR5 (SOX2^+^LGR5^low^ cells) were shown as arrows (**b’**,**d’**,**f’**). Bar = 50 µm ((**a**–**f**) are the same sections) or 25 µm (**b’**,**d’**,**f’**). DSS, dextran sodium sulfate; LGR5, leucine-rich repeat-containing G-protein-coupled receptor 5; SOX2, sex-determining region on Y-box transcription factor 2; SOX9, sex-determining region on Y-box transcription factor 9; TZ, transitional zone.

**Table 1 ijms-25-12706-t001:** Expression of cytokeratin in the transitional zone.

				Sample 1		Sample 2		Sample 3		Mean	
Group	Keratin Type		Accession	Score	emPAI	Score	emPAI	Score	emPAI	Score	emPAI
DSS	Type I	CK10	A2A513	ND	ND	ND	ND	1482	1.48	(1482)	(1.48)
		CK13	P08730	31	0.14	138	0.29	376	0.77	545	1.2
		CK14	Q61781	179	0.5	ND	ND	ND	ND	(179)	(0.5)
		CK42	Q6IFX2	ND	ND	ND	ND	131	0.2	(131)	(0.2)
	Type II	CK1	P04104	ND	ND	ND	ND	1466	1.2	(1466)	(1.2)
		CK4	P07744	ND	ND	ND	ND	356	0.46	(356)	(0.46)
		CK5	Q922U2	194	0.41	399	0.56	156	0.28	645	1.25
		CK6a	P50446	147	0.29	368	0.43	ND	ND	331	0.72
		CK6b	O3UV11	ND	ND	ND	ND	ND	ND	-	-
		CK8	P11679	ND	ND	ND	ND	140	0.18	(140)	(0.18)
		CK15	B1AQ77	ND	ND	187	0.2	ND	ND	(187)	(0.2)
		CK73	Q6NXH9	ND	ND	ND	ND	ND	ND	-	-
Control	Type I	CK10	A2A513	90	0.11	ND	ND	ND	ND	(90)	(0.11)
		CK13	P08730	ND	ND	ND	ND	151	0.46	(151)	(0.46)
		CK14	Q61781	ND	ND	ND	ND	ND	ND	-	-
		CK42	Q6IFX2	ND	ND	ND	ND	ND	ND	-	-
	Type II	CK1	P04104	52	0.1	ND	ND	ND	ND	(52)	(0.1)
		CK4	P07744	ND	ND	ND	ND	ND	ND	-	-
		CK5	Q922U2	ND	ND	ND	ND	78	0.22	(78)	(0.22)
		CK6a	P50446	ND	ND	ND	ND	139	0.29	(139)	(0.29)
		CK6b	O3UV11	ND	ND	46	0.11	ND	ND	(46)	(0.11)
		CK8	P11679	ND	ND	ND	ND	ND	ND	-	-
		CK15	B1AQ77	ND	ND	103	0.06	20	0.13	123	0.19
		CK73	Q6NXH9	ND	ND	ND	ND	33	0.05	(33)	(0.05)

Abbreviation: CK, Cytokeratin; DSS, dextran sulfate sodium; emPAI, Exponentially modified protein abundance index; ND, Not detected. The number of parentheses represents one sample.

## Data Availability

The original contributions in this study are included in the article. Further inquiries can be directed to the corresponding authors.

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
