# Peer review of "Anorectal Remodeling in the Transitional Zone with Increased Expression of LGR5, SOX9, SOX2, and Keratin 13 and 5 in a Dextran Sodium Sulfate-Induced Mouse Model of Ulcerative Colitis"

_ijms, 2024, doi:10.3390/ijms252312706_

Round 1
Reviewer 1 Report
Comments and Suggestions for Authors
ijms-3322316
The manuscript by Kobayashi et al. reports interesting findings that demonstrate the mechanistic insights of abnormal repair of the anorectal region in mice treated with DSS. They showed that abnormal keratin assembly and stem cell activation interfere with rectal crypt regeneration, leading to pathological anorectal remodeling in severe colitis. The animal experiments were carefully performed and too many data were obtained. Although the manuscript is well written, this reviewer would like to suggest several points listed below for publication.
(1) Please define the word “excessive hyperplasia”. What does it mean? Also, please explain or rephrase “abnormal repair of the anorectal region, which consisted of mixed pseudocarcinomatous hyperplastic TZ and regenerating crypts” for readers understanding.
(2) Are there pathologic findings obtained by this experiment in the anorectal region of UC patients?
(3) Please describe the clinical significance of results obtained by this experiment.
(4) The authors focused the anorectal region in mice exposed to DSS. I think DSS may affect lower parts of the colon, including anorectal region. Please shortly describe the reason why DSS affects a limited colonic and anorectal regions.
Author Response
(1) Please define the word “excessive hyperplasia”. What does it mean? Also, please explain or rephrase “abnormal repair of the anorectal region, which consisted of mixed pseudocarcinomatous hyperplastic TZ and regenerating crypts” for readers understanding.
(Response)
Thank you for your insightful comments.
In our ongoing studies using the DSS colitis model, we were initially surprised to observe TZ hyperplasia in the distal colon and faced challenges in distinguishing it from neoplastic change. Several reports supported the diagnosis (Mcnairn et al., 2011; Perše et al., 2012; Mitoyan et al., 2021; Liu et al., 2022; Sugimoto et al., 2022). Consequently, we have used terms like “excessive hyperplasia” and “pseudocarcinomatous hyperplasia”to describe the findings.
Excessive hyperplasia refers to severe hyperplasia, characterized by abnormal, but non-neoplastic, proliferation of epithelial cells.
Pseudocarcinomatous hyperplasia or pseudoepitheliomatous hyperplasia, mimics squamous cell carcinoma (SCC) but is a reactive, non-malignant process. This type of hyperplasia often originates from adnexal structures (e.g., follicular infundibula, eccrine ducts (Grunwald et al., 1988; Nayak et al., 2015) rather than epidermal epithelium. It can occur as a response to inflammatory or neoplastic conditions, such as deep fungal infections or lymphomas (Wipf et al., 2019; Jeunon et al., 2020).
In this study, to avoid misunderstanding, we concluded that TZ changes represent hyperplasia rather than neoplasia and have described them as “hyperplastic TZ.”
To clarify, we have revised “abnormal repair of the anorectal region, which consisted of mixed pseudocarcinomatous hyperplastic TZ and regenerating crypts” to:
“abnormal repair of the anorectal region, which consisted of mixed hyperplastic TZ and regenerating crypts.”
References:
Mitoyan L, et al. Nat. Commun. 2021, 12, 2761.
Liu CY, et al. Gastroenterology 2022, 162, 1975-1989.
Perše M and Cerar A. J. Biomed. Biotechnol. 2012, 2012, 718617.
Mcnairn AJ and Guasch G. Eur. J. Dermatol. 2011, 21 Suppl 2, 21-8.
Sugimoto S, et al. Gastroenterology 2022, 162, 334-337.e5.
Jeunon T, et al. Am J Dermatopathol. 2020;42(9):662-672.
Grunwald MH, et al. Am J Dermatopathol. 1988;10(2):95-103.
Wipf A, et al. J Cutan Pathol. 2019;46(7):520-527.
Nayak VN, et al. J Int Oral Health. 2015 Sep;7(9):148-52.
(2) Are there pathologic findings obtained by this experiment in the anorectal region of UC patients?
(Response) Thank you for your question.
As discussed in the 4th paragraph of the DISCUSSION section, similar findings have been reported in patients with ulcerative colitis-associated neoplasm (UCAN). Sugimoto et al. documented squamous metaplasia, in which non-squamous epithelium in the distal rectum of long-term UC patients, where non-squamous epithelium was replaced by stratified squamous epithelium. Endoscopic observations of UCAN have also identified squamous metaplasia as a histopathologic finding. Similarly, Sugimoto et al. documented squamous epithelization in a mouse model using the EDTA-breaching technique, where squamous epithelia replaced mucosal regions during wound healing. These findings align with squamous epithelialization observed in our DSS model.
Regarding the link between squamous metaplasia and tumor development, Cheng et al reported SCC arising from squamous metaplasia in a UC patient with longstanding disease and perianal warts. This suggests a potential progression from squamous metaplasia to neoplastic transformation under specific conditions.
References:
Sugimoto S, et al., Gastroenterology. 2022;162(1):334-337.e5.
Cheng H, et al., Can J Gastroenterol. 2007;21(1):47-50.
(3)Please describe the clinical significance of results obtained by this experiment.
(Response) Thank you for question.
The clinical signidicance of the results was addressed by representing the DAI, which we previously reported in Experiment I (Kobayashi et al., 2022). The DAI for Experiments II-IV ares shown in revised sFig. 7-9, respectively.
References:
Kobayashi M, et al., Dig Dis Sci. 2022;67(10):4770-4779.
(4)The authors focused the anorectal region in mice exposed to DSS. I think DSS may affect lower parts of the colon, including anorectal region. Please shortly describe the reason why DSS affects a limited colonic and anorectal regions.
(Response) Thank you for your insightful comments. We demonstrated that DSS primarily affects the distal colon and anorectal region in mice. Okayasu et al. first described this phenomenon, where DSS-induced colitis caused diarrhea, rectal bleeding, and weight loss with lesions primarily localized to the left (distal) colon. DSS with a molecular weight of ~40 kDa induces the most severe colitis, particularly in BALB/c mice, affecting the middle and distal colon (Kitajima et al., 2000). Its toxicity disrupts colonic epithelial integrity, increases permeability, and creates a microenvironment favorable for bacterial overgrowth, which contributes to localized inflammation in the distal colon (Hans et al., 2000).
Mouse strain also influences DSS-induced colitis. For example, Balb/c mice develops severe distal coltis, while other strains like C3H show mainly in the proximately colon and cecum (Stevceva et al. 1999).
In this experiment, the use of 40 kDa DSS and Balb/c mice likely explains the limited effects in the distal colon and anorectal regions.
References:
Okayasu I, et al., Gastroenterology. 1990;98(3):694–702.
Stevceva L, et al., J Gastroenterol Hepatol. 1999;14(1):54-60. .
Kitajima S, et al.,Exp Anim. 2000;49(1):9-15.
Chassaing B, et al., Curr Protoc Immunol. 2014;104:15.25.1-15.25.14.
Hans W, et al., Eur J Gastroenterol Hepatol. 2000;12(3):267-73.
Reviewer 2 Report
Comments and Suggestions for Authors
The manuscript entitled „Anorectal remodeling in the transitional zone with increased expression of LGR5, SOX9, SOX2, and keratin 13 and 5 in a dextran sodium sulfate-induced mouse model of ulcerative colitis” aimed to verify the keratin subtypes and examined the expression of stem cell markers in anorectal transitional zone.
The introduction is informative and properly introduce the problem. However, the authors could justify why they only used females in the study.
The methods are well described and repeatable. The authors could add the ethical committee number for performed in vivo studies.
The results are presented in tables and graphs which are clear and on good quality. Also the results are described in a logical way and convey the main observations without exaggeration.
In the discussion part the authors confront their results with others. The authors should underline that their results referred to females and if they are other studies on this sex or if they expect contradictory or similar results on males. Additionally, the authors included young female mice and this type of carcinoma is more often in older population. Do the authors plan to perform similar studies on older mice (e.g. 6 months old females which oestrogen level are after puberty time and the changed level of oestrogen may change the results and conclusions)?
Author Response
(1) The introduction is informative and properly introduce the problem. However, the authors could justify why they only used females in the study.
(Response) Thank you for highlighting this point.
We selected female Balb/c mice for this study due to their suitability for analyzing mucosal regeneration in refractory ulcerative lesions, particularly in the left-sided and distal colon, which affected two-thirds of human ulcerative colitis cases. Balb/c mice are known to develop severe distal colitis (Stevceva et al., 1999) and recover more effectively than C57BL/6 mice, making them ideal for studying mucosal repair during the acute phase (Melgar et al., 2005).
We also chose females for two other reasons. First, females develop colitis less aggressively than males, which aligns with animal welfare considerations. Second, male Balb/c mice exhibit higher aggression, which may relate to stress that may exacerbate colitis severity. These factors make female Balb/c mice a suitable model for our study objectives (Hoffmann et al., 2017).
References:
Stevceva L, et al., J Gastroenterol Hepatol. 1999;14(1):54-60.
Melgar S, et al., Am J Physiol Gastrointest Liver Physiol. 2005;288(6):G1328-38.
Hoffmann M, et al., Lab Anim. 2018;52(3):240-252.
(2) The methods are well described and repeatable. The authors could add the ethical committee number for performed in vivo studies.
(Response)
Thank you for your comment. We apologize for omitting the ethical committee approval numbers for the animal studies. The approval numbers for Experiments I-IV are as follows: No. 31-67, 31-78, R04-133, and R06-163, respectively.
(3) The results are presented in tables and graphs which are clear and on good quality. Also the results are described in a logical way and convey the main observations without exaggeration.
(Response)
Thank you very much for your kind review and comments.
(4) In the discussion part the authors confront their results with others. The authors should underline that their results referred to females and if they are other studies on this sex or if they expect contradictory or similar results on males. Additionally, the authors included young female mice and this type of carcinoma is more often in older population. Do the authors plan to perform similar studies on older mice (e.g. 6 months old females which oestrogen level are after puberty time and the changed level of oestrogen may change the results and conclusions)?
(Response)
In mouse models of ulcerative colitis, it has been demonstrated that estrogen receptor (ER) activation in female exacerbates acute colitis, while ERα inhibition reduces inflammation. This suggests that ovarian hormones, particularly, play a significant role in driving nflammation (Hjelt et al., 2024). However, studies specifically investigating the role of ER signaling in TZ hyperplasia remain limited. Existing research on TZ hyperplasia have either included male and female mice (Mitoyan et al., 2021; Liu et al., 2022) or have not expilicitly reported sex-specific differences (Perše et al., 2012; Mcnairn et al., 2011; Sugimoto et al. ., 2022). As such, the influence of sex on TZ hyperplasia and its response to estrogen signaling remains unclear.
Our study focused exclusively on young female mice, as they are more resilient to severe inflammation compared to males. However, we recognize that colorectal carcinomas are more commonly observed in older populations, and hormonal changes associated with aging may significantly alter inflammatory and regenerative responses. We agree that future studies should investigate the role of estrogen in TZ hyperplasia through detailed comparisons of sexes and age groups, including young-adult and post-reproductive (retired breeder) female mice, particularly with inactivation of estrogen signaling. These studies will provide critical insights into the interplay between estrogen signaling, age, and TZ hyperplasia, potentially enhancing the translational relevance of the model.
References:
Mitoyan L, et al. Nat. Commun. 2021, 12, 2761.
Liu CY, et al. Gastroenterology 2022, 162, 1975-1989.
Perše M, and Cerar A. J. Biomed. Biotechnol. 2012, 2012, 718617.
Mcnairn AJ, and Guasch G. Eur. J. Dermatol. 2011, 21 Suppl 2, 21-8.
Sugimoto S, et al. Gastroenterology 2022, 162, 334-337.e5.
Round 2
Reviewer 2 Report
Comments and Suggestions for Authors
The authors addressed all my comments.